# Long-Term Effectiveness of a Smartphone App and a Smart Band on Arterial Stiffness and Central Hemodynamic Parameters in a Population with Overweight and Obesity (Evident 3 Study): Randomised Controlled Trial

**DOI:** 10.3390/nu14224758

**Published:** 2022-11-10

**Authors:** Leticia Gómez-Sánchez, Marta Gómez-Sánchez, Cristina Lugones-Sánchez, Emiliano Rodríguez-Sánchez, Olaya Tamayo-Morales, Susana Gonzalez-Sánchez, Rosa Magallón-Botaya, Jose Ignacio Ramirez-Manent, Jose I. Recio-Rodriguez, Cristina Agudo-Conde, Luis García-Ortiz, Manuel A. Gómez-Marcos

**Affiliations:** 1Unidad de Investigación en Atención Primaria de Salamanca (APISAL), Gerencia de Atención Primaria de Salamanca, Gerencia Regional de salud de Castilla y León (SACyL), Avenida de Portugal 83, 37005 Salamanca, Spain; 2Instituto de Investigación Biomédica de Salamanca (IBSAL). Paseo de San Vicente 58-182, 37007 Salamanca, Spain; 3Red de Investigación en Cronicidad, Atención Primaria y Promoción de la Salud (RICAPPS) (RD21/0016), 08007 Barcelona, Spain; 4Departamento de Medicina, Universidad de Salamanca, Calle Alfonso X el Sabio s/n, 37007 Salamanca, Spain; 5Centro de Salud Arrabal, Grupo Aragonés de Investigación en Atención Primaria, Instituto de Investigaciones Sanitarias de Aragón (IIS), Servicio de Salud de Aragón, 50002 Zaragoza, Spain; 6Departamento de Medicina y Dermatología. Universidad de Zaragoza, 50009 Zaragoza, Spain; 7Centro de salud de Calvià, Instituto de Investigación Sanitaria de Baleares (IDISBA), Servicio de Salud de las Islas Baleares, 07180 Mallorca, Spain; 8Departamento de Medicina Universidad de las Islas Baleares, 07003 Islas Baleares, Spain; 9Departamento de Biomedicina y Diagnostico de la Ciencia, Universidad de Salamanca, Calle Alfonso X el Sabio s/n, 37007 Salamanca, Spain

**Keywords:** mobile app, telemedicine, eHealth, weight control, arterial stiffness, central hemodynamic parameters, mobile phone

## Abstract

Background: mHealth technologies could help to improve cardiovascular health; however, their effect on arterial stiffness and hemodynamic parameters has not been explored to date. Objective: To evaluate the effect of a mHealth intervention, at 3 and 12 months, on arterial stiffness and central hemodynamic parameters in a sedentary population with overweight and obesity. Methods: Randomised controlled clinical trial (Evident 3 study). 253 subjects were included: 127 in the intervention group (IG) and 126 in the control group (CG). The IG subjects were briefed on the use of the Evident 3 app and a smart band (Mi Band 2, Xiaomi) for 3 months to promote healthy lifestyles. All measurements were recorded in the baseline visit and at 3 and 12 months. The carotid-femoral pulse wave velocity (cfPWV) and the central hemodynamic parameters were measured using a SphigmoCor System^®^ device, whereas the brachial-ankle pulse wave velocity (baPWV) and the Cardio Ankle Vascular Index (CAVI) were measured using a VaSera VS-2000^®^ device. Results: Of the 253 subjects who attended the initial visit, 237 (93.7%) completed the visit at 3 months of the intervention, and 217 (85.3%) completed the visit at 12 months of the intervention. At 12 months, IG showed a decrease in peripheral augmentation index (PAIx) (−3.60; 95% CI −7.22 to −0.00) and ejection duration (ED) (−0.82; 95% CI −1.36 to −0.27), and an increase in subendocardial viability ratio (SEVR) (5.31; 95% CI 1.18 to 9.44). In CG, cfPWV decreased at 3 months (−0.28 m/s; 95% CI −0.54 to −0.02) and at 12 months (−0.30 m/s, 95% CI −0.54 to −0.05), central diastolic pressure (cDBP) decreased at 12 months (−1.64 mm/Hg; 95% CI −3.19 to −0.10). When comparing the groups we found no differences between any variables analyzed. Conclusions: In sedentary adults with overweight or obesity, the multicomponent intervention (Smartphone app and an activity-tracking band) for 3 months did not modify arterial stiffness or the central hemodynamic parameters, with respect to the control group. However, at 12 months, CG presented a decrease of cfPWV and cDBP, whereas IG showed a decrease of PAIx and ED and an increase of SEVR.

## 1. Introduction

Obesity and sedentariness are among the main health problems in developed countries [1,2]. Therefore, in the last decades, different multicomponent interventions have been implemented to address different lifestyles, showing promising results in the prevention of unhealthy lifestyles [3,4]. Such interventions include those focused on increasing physical activity and reducing the intake of calories [5,6].

Arterial stiffness, measured in a non-invasive manner with carotid-femoral pulse wave velocity (cfPWV), brachial-ankle pulse wave velocity (baPWV), and Cardio-ankle vascular index (CAVI), has shown a positive association with the number of cardiovascular events [7,8,9,10,11]. cfPWV, measured through tonometry, is considered the reference measurement and reflects the arterial stiffness of the descending aorta, iliac arteries, first part of the femoral arteries, and common carotid artery, but it does not evaluate the ascending aorta [12]. baPWV measures peripheral artery stiffness through the analysis of the tibial and brachial artery waves [13]. Lastly, the cardio-ankle vascular index (CAVI), evaluated through oscillometry, analyses the stiffness of the aortic (including the ascending aorta), iliac, femoral and tibial arteries, and it does not depend on arterial pressure at the time when the measurement is recorded [14].

Central arterial pressure measured with non-invasive techniques is gaining relevance, as it directly affects the heart, kidneys and brain [15], and it predicts cardiovascular events more accurately than peripheral arterial pressure [16]. Moreover, the study of the parameters derived from the forward pulse wave, generated by the cardiac systole, and those derived from the retrograde pulse wave, generated in the arterial branch points, provides information about the vascular tree [17]. Such parameters include the central augmentation index (CAIx)—which is the most used substitute to analyse the reflection of the arterial wave–, the peripheral augmentation index (PAIx), augmentation pressure (AP), ejection duration (ED) and subendocardial viability ratio (SEVR) [18]. These central hemodynamic parameters show greater association with morbimortality by cardiovascular diseases [19] than peripheral arterial pressure, and they present a different response of antihypertensive drugs [20,21].

Since the last decade, the interventions to improve health based on new technologies are acquiring relevance, appearing as promising tools due to their growth capacity, low cost and potential use in different scenarios [22]. The use of mobile health (mHealth) could optimise these efforts, improve the follow-up, and help the patient in his/her health management [23,24]. The mHealth interventions that include mobile phones have shown greater efficacy than traditional interventions to lose weight [25,26], although the improvements obtained to increase physical activity are lower [27]. In this sense, activity-tracking bands have proven valid and reliable to measure the results of physical activity [28]. Therefore, it is necessary to design multicomponent strategies that can favour the attainment of positive results in diet, physical activity and other health variables [29,30,31]. Taking into account all of the above mentioned, the intervention of the EVIDENT 3 study was designed [32].

This document presents secondary results of the EVIDENT 3 research project [32], whose main objectives were to evaluate the long-term effectiveness (12 months) of a multicomponent intervention on weight loss and increase of physical activity in Spanish sedentary adults with overweight or obesity.

Taking into account that the effects of an mHealth-based multicomponent intervention on arterial stiffness, central blood pressure and hemodynamic parameters have not been studied in sedentary patients with overweight or obesity, we set out to analyze whether an intervention such as the one developed in the EVIDENT 3 [5] project would have an effect on these parameters. In addition, it is not clear how long this type of intervention needs to last to be effective, nor if its effects are maintained over time.

The objective of this study is to evaluate the short-term (3 months) and long-term (12 months) efficacy of a multicomponent mHealth intervention, comprising the version with the smartphone application (EVIDENT 3 study) and a smart band, compared to the brief advice-only intervention on arterial stiffness, central blood pressure, and pulse wave-derived parameters in sedentary overweight and obese Spanish adults.

## 2. Materials and Methods

### 2.1. Study Design

A multicentre, randomised, controlled, clinical trial was conducted with 2 parallel groups in a context of primary care (EVIDENT 3). The individuals included in this manuscript were those recruited in the Primary Care Research Unit of Salamanca (APISAL), which coordinates the EVIDENT 3 study [32] (ClinicalTrials.gov NCT03175614), since it is the subgroup that was subjected to a complete analysis of arterial stiffness and the parameters derived from the pulse wave. Between June 2017 and June 2020, evaluations were carried out at the beginning of the study, and at 3 and 12 months. The results presented in this manuscript correspond to the 253 individuals included in the Salamanca group.

### 2.2. Study Population

The subjects were selected by simple random sampling among the patients who visited their primary care doctor. In this essay we have used as inclusion criteria, subjects aged between 20 and 65 years, body mass index (BMI) between 27.5 kg/m^2^ and 40 kg/m^2^, and being classified as sedentary (less than 20 min of vigorous physical activity ≤3 times per week; less than 30 min of moderate physical activity ≤5 times per week; or a combination of moderate and vigorous physical activity ≤5 times per week) [33].

The sample size was calculated for a measurement of arterial stiffness, i.e., cfPWV, as the main variable of this study. Establishing an alpha and beta risk of 0.05 and 0.20, respectively, with SD = 2.03 m/s, estimated in the subjects of the EVA study [34], and assuming a correlation between measurements of 0.8, a total of 122 individuals would be required to detect a cfPWV difference of ≥0.50 m/s between the intervention group (IG) and the control group (CG), assuming a 15% loss to follow-up.

### 2.3. Approval of the Ethics Committee and Consent of the Subjects to Participate in the Trial

The study was approved in April 2016 by the Drug Research Ethics Committee of Salamanca. All procedures were carried out following the recommendations of the Declaration of Helsinki [35]. All subjects signed the written informed consent document before participating in the study. The study was registered in ClinicalTrials.gov (registration n° number: NCT03175614) on 31 May 2017.

### 2.4. Randomisation

The subjects of this study were randomised to IG or CG after attending the initial visit and signing the informed consent. The randomisation sequence was generated with a 1:1 proportion, using the Epidat software package (version 4.2; Xunta de Galicia, Spain) [36], by an independent researcher, who was blinded to the group randomisation. Due to the nature of the study, the individuals could not be blinded to the intervention.

### 2.5. Procedures

Each participant had an initial visit and two follow-up visits: one at 3 months and another at 12 months (Figure 1). None of the participants received compensation for participating in the study or for attending visits during the study.

The data of the visits were compiled by a researcher nurse, who completed the data gathering notebook in paper format. Subsequently, the data were registered electronically on the EVIDENT 3 study website. An additional visit 7 days after the initial one was programmed for the IG participants to brief them on the use of the mobile application and the activity-tracking smart band, as well as on how to set up both devices with their own data. The researcher who conducted the additional visit was different from the researcher who collected the data of the initial visit and in the follow-up visits at 3 and 12 months.

### 2.6. Primary Results

The primary objective of this study was to decrease arterial stiffness evaluated with cfPWV, baPWV and CAVI. Moreover, we also analysed the changes in central arterial pressure and in the different parameters derived from the analysis of the pulse wave.

All measurements were taken between 8 and 10 h in the morning, in consultation without noise and with a temperature between 22–28 °C, fasting for at least 8 h, without having taken any medication since the night before and without having smoked or done physical exercise at least 1 h before. Arterial stiffness measurements were performed with the patients in the supine position. The pulse wave study was performed with the participant sitting.

### 2.7. Measurement of Arterial Stiffness

cfPWV was measured using a SphygmoCor device (AtCor Medical Pty Ltd., Head Office, West Ryde, Australia). Following the recommendations established by Van Borte et al. [12], we measured the distance between the sternal notch and the point where we placed the sensor in the carotid artery and in the femoral artery with a tape measure.

CAVI and ba-PWV were measured using a VaSera VS-2000^®^ device (FukudaDenshi, Denshi Co. Ltd., Tokyo, Japan), following the manufacturer’s instructions. The muffs were selected based on the arm and leg circumference of each participant. We also used a 2-sided adhesive tape to place a heart sound microphone on the sternum, at the level of the second intercostal space. CAVI measurements are considered valid when they are obtained for at least 3 consecutive heartbeats [37]. The CAVI values were automatically calculated by replacing the stiffness parameter in the following equation: stiffness parameter β = 2ρ × 1/(systolic arterial pressure-diastolic arterial pressure in mmHg) × ln (systolic arterial pressure/diastolic arterial pressure in mmHg) × PWV2, where ρ is blood density [3]. baPWV was estimated using the following equation: baPWV = (0.5934 × height (cm) + 14.4724)/time interval between the arm and ankle waves [37].

### 2.8. Measurement of the Central Arterial Pressures and Analysis of the Pulse Wave

Through tonometry, placing a sensor in the radial artery and through a mathematical transformation, based on the publication by Takazawa K et al. [38], which the manufacturer has incorporated into the SphygmoCor^®^ System device (AtCor Medica lPty Ltd., West Ryde, Australia), the CSBP and CDBP as well as the aortic pulse wave were estimated. The following measurements were made: Systolic (cSBP) and diastolic (cDBP) central arterial pressures. Pulse Pressure (PA) magnification calculated as an absolute value (Peripheral Pulse Pressure-Central Pulse Pressure). Subendocardial viability index (SEVR %) estimated with the equation SEVR % = wave area in diastole/wave area during systole and ejection duration (ED) in (%) (ED = ED of systole/total duration of the cardiac cycle * 100).

### 2.9. Adherence to the Smartphone App

Adherence to the application was calculated from the number of days since the participants started the session in the application and recorded their first meal. The number of days that the subjects made records in the App were classified into four categories: 0 days, between 1 and 30 days, between 31 and 60 days and more than 60 days. Subjects who used the app for more than 60 days were classified as good adherence, while ≤60 days of use were classified as low adherence.

### 2.10. Other Variables

In the initial visit, the age and sex of the subjects were included among the sociodemographic variables. The measurement of the cardiovascular risk factors has been previously published in detail in the study protocol [32].

### 2.11. Intervention

The detailed content of the brief advice, given to both CG and IG, and of the specific intervention carried out in IG is described in the study protocol [32]. All documents used during the intervention were provided in Spanish.

#### 2.11.1. Standard Advice Conducted in the Two Groups

The brief advice was performed before the randomisation conducted by the nurse, who was trained for this study and did not participate in other aspects of it. The brief advice consisted in 5-min recommendations about lifestyles, focusing on the benefits of physical activity and cardiohealthy diet. The health benefits of physical activity were explained, as well as the recommendation of carrying out at least 30 min of moderate physical activity 5 days per week, or 20 min of vigorous physical activity 3 days per week. The advice about eating was conducted following the plate method [39], where the plate is divided into 4 parts: half of the plate must be salad or vegetables, a fourth part of the plate must be proteins (preferably white meat over red meat), and another fourth part of the plate must be carbohydrates. Moreover, for dessert, the subjects were requested to have a medium-sized piece of fruit and a fat-free dairy product. No counselling support was provided during the study.

#### 2.11.2. Specific Intervention of IG

In addition to the common intervention carried out in both groups, a 15-min scheduled visit was made to the IG seven days after the initial visit, in which a mobile phone with the EVIDENT 3 application (Samsung Galaxy, Samsung, Suwon, Korea) and a smart band (Xiaomi Miband 2, Xiaomi, Beijing, China) were provided, to be used during the 3 months of the intervention, and to explain the use of the application (EVIDENT 3 app [Registration no. 00/2017/2438]) designed for this study by CGB Computer Company and APISAL, as well as the operation of the smart band. Also on this visit, the app was configured with each participant’s data (sex, age, weight and height) to establish the individual goals that they should achieve. The application was designed to perform self-registration of diet data.

The composition of the different foods ingested in the diet and dishes present in the app were obtained from the Spanish food composition database, developed by the Spanish Network of Food Composition Databases and by the Spanish Agency for Food Safety and Nutrition. In the app, each participant entered a daily record of their intake from each of the meals made, selecting from among the dishes and foods available in it and portion size. In case the eaten dish did not appear, they were instructed to choose the one with the most similar characteristics. After all daily information has been collected, the application integrates and analyzes all the data to generate personalized recommendations. The app displays the total calories recorded daily (Appendix A) and a bar that changes color based on the set values (green, yellow, or red). The application was configured for a hypocaloric diet, estimating the upper range (the red line), according to age and sex, basal metabolic rate, diet-induced thermogenesis and estimated energy expenditure for sedentary activities. The lower limit (the black line) reflects 85% of the calculated calories, and below this, the bar appears in green; between the red and black lines, the bar is yellow, and above the red line, it is red. At the end of each day, the user can consult the application for recommendations and information on changes in caloric intake and the distribution of macronutrients (carbohydrates, proteins, and saturated and unsaturated fats) (Appendix A). Physicians, nurses, dieticians and psychologists collaborated in the development of the application and effective behavioral strategies were taken into account to achieve changes in habits, such as self-registration [40], goal setting and positive reinforcement [41].

The physical activity data collected by the smart band was synchronized with the app. Similarly, congratulation messages were generated by the activity data when 10,000 steps/day were reached. The participant could consult this information in the application.

At the 3-month visit, participants returned the intervention-specific devices to the researchers. From that moment, the users had no further access to the mobile phone or the activity recording smart band used in the app during the intervention, and they were also instructed not to use other digital tools aimed at losing weight until the end of the trial. All data generated by the EVIDENT 3 app were analyzed and entered into the EVIDENT study website.

### 2.12. Blinding Strategy

The specific intervention in IG was carried out by a different researcher, who was not involved in the evaluation, data analysis or the standard advice. The researchers were blinded throughout the entire study. Due to the study design, the subjects could not be blinded. In the two follow-up visits carried out, at 3 and 12 months, only the study variables were evaluated, without any counseling or reinforcement, in order to avoid contamination between the two groups. The app was not available for download on the internet or elsewhere until the end of the study, so CG was unable to use it. Additionally, at follow-up visits, included subjects were expressly asked not to use other digital health tools 2.13. Statistical analysis

The analysed quantitative variables of the study population were reflected as mean ± standard deviation, and the qualitative variables were reflected as number and percentage. To analyse the differences between IG and CG in the baseline visit, Student’s t-test and chi-squared test were used. All analyzes were done by intention to treat Paired Student’s t-test was applied to analyse the intragroup changes, and Independent Student’s t-test was employed for the intergroup differences. To analyse the effect of the intervention, we conducted repeated-measures analyses of variance, using the general linear model, comparing the changes observed between IG and CG in the analysed variables. The hypothesis test permanent an alpha value of 0.05. The statistical program used to analyze was SPSS Statistics software (version 25.0; IBM Corporation, Armonk, New York, USA).

## 3. Results

### 3.1. Baseline Characteristics of the Subjects and Follow-up

Of the 253 subjects who attended the initial visit, 237 (93.7%) completed the follow-up visit at 3 months, and 217 (85.3%) completed the follow-up visit at 12 months. In the flowchart of the study, it is described how the selection, inclusion and follow-up of the subjects were carried out. (Figure 1).

The clinical and sociodemographic characteristics at the beginning of the study of the 253 (IG = 127, CG = 126) subjects included are shown in Table 1. The subjects had a mean age of 47.9 ± 9.8 years (69.6% women). A total of 27% presented hypertension, 21.4% had dyslipemia, and 21% were smokers. In the initial evaluation, there were no differences in any of the studied variables between the two groups.

No differences were found in any of the initial characteristics between the 36 individuals (13 in IG and 23 in CG) who left the study and those who completed it (Appendix A).

### 3.2. Adherence to Self-Monitoring on the Smartphone App

The median application use was 76 days of the three months intervention period, with no differences being detected between sexes (men: 74 days vs. women: 77 days). Adherence according to the number of days that the App was used is represented in Appendix A, which shows that 65% of the individuals used the App for over 60 days.

In the correlation analysis between the days of use of the App and the changes in the measurements of arterial stiffness at 3 and 12 months, we found a negative correlation with all of such measurements, although significance was only reached with the change at 3 months in baPWV (r = −0.180; *p* = 0.047) and cDBP (r = −0.206; *p* = 0.023). No correlation was identified with any parameter of the pulse wave. The results are present in Appendix A globally and by sex.

### 3.3. Changes in the Measurements of Stiffness Analysed throughout the Study Period

Table 2 shows the baseline values, the 3-month and 12-month differences of the intragroup stiffness measurements, and the effect of the intervention (differences between IG and CG). None of the analysed measurements of arterial stiffness showed differences between the two groups, neither in the baseline visit nor in the follow-up visits (*p* > 0.05). In the intragroup analysis, cfPWV decreased at 3 and 12 months with respect to the baseline evaluation in the two groups, although the differences were significant only in CG, at 3 and 12 months. No intragroup differences were found with baPWV or CAVI, except in CG in the 12-month evaluation, where CAVI increased.

Figure 2 shows the evolution of the measurements of stiffness analysed throughout the study. cfPWV decreased at 3 months in the two groups, although the decreasing tendency remained in CG at 12 months, while this parameter increased in IG. baPWV and CAVI increased at 12 months in the two groups, presenting a greater increasing tendency in CG.

### 3.4. Changes in the Measures of Central Pressures and Parameters Derived from the Analysis of the Pulse Wave throughout the Study

Table 3 shows the baseline values, the 3-month and 12-month intragroup differences of the central arterial pressures and pulse wave measurements, and the effect of the intervention (differences between IG and CG). None of the analyzed measurements showed differences between the GI and the CG at the initial visit, 3-month or 12-month visit (*p* > 0.05). In the intragroup analysis, cSBP and cDBP decreased at 3 and 12 months with respect to the initial evaluation in the two groups, although statistical significance was only obtained in cSBP at 3 months and in cDBP at 12 months in the CG. In the pulse wave analisis, only the following measurements presented differences in IG with respect to their baseline values: CAIx and ED decreased at 12 months and SEVR increased at 12 months.

Figure 3 presents the evolution of the central arterial pressures, CAIx and PAIx. All these parameters decreased at 3 months in the two groups, and either increased or stabilised at 12 months.

Figure 4 displays the evolution of AP, ED and SEVR. AP increased at 3 and 12 months in IG, whereas in CG it increased at 3 months and decreased at 12 months. ED decreased at 3 and 12 months in both groups. SEVR increased in IG at 3 and 12 months, whereas in CG it decreased at 3 months and increased at 12 months.

## 4. Discussion

In this study conducted in sedentary overweight or obese subjects, brief lifestyle advice combined with the use of the tool designed in the EVIDENT 3 study had no effect on the measures of arterial stiffness, central arterial pressures, and central hemodynamic parameters analyzed, compared to the intervention with only brief lifestyle advice. In the CG, however, cfPWV decreased at 3 and 12 months and CAVI increased at 12 months. In the IG, cSBP decreased at 3 months and cDBP, CAIx and ED at 12 months, and SEVR increased at 12 months. These results suggest that brief advice recommending heart-healthy lifestyles, based on diet and physical activity, may be useful to improve the evolution of several of the parameters analyzed in this study, implying that the effect of multicomponent interventions may be small or even nonexistent. Such results can be explained in different ways, by the low intensity of the intervention, short period of use of the app, or that the effect of brief advice could be enhanced in the CG because they feel observed.

These arguments are supported by the results of the recently published meta-analysis, which concludes that frequent and sustained interventions are needed to achieve clinically significant weight loss of 5% [42] publicado recientemente, que concluye que se necesitan intervenciones frecuentes y sostenidas para lograr una pérdida de peso clínicamente significativa del 5%.

On the other hand, when proposing this type of studies, it is important to consider that most of the parameters explored in this work are modified not only with arterial pressure, but also with age [43,44,45]. Thus, in the EVA study, which analysed several of these measurements in a Spanish population without cardiovascular diseases, found that, after controlling for the mean arterial pressure, the yearly increase of these parameters was: cfPWV 0.080 m/s 95%CI (0.071–0.090), baPWV 0.119 m/s 95%CI (0.109–0.130), CAVI 0.069 95%CI (0.063–0.075), and CAIx 0.410 95%CI (0.337–0.493) [34]. Likewise, the values of cSBP and cDBP increase with age, and those of SEVR decrease, showing no differences regarding age in ED or AP [46]. Therefore, when interpreting the decreases obtained in several of these measurements, it is convenient to take into account the increases that are due to ageing, which, presumably, would have occurred if the intervention with the brief advice had not been carried out in both groups.

The trend observed at three months towards decreases in values such as cfPWV and CAIx, and the disappearance of this trend at 12 months, has been observed previously in the results published in the EVIDENT 2 study [47]. These results are also consistent with those found in several publications generated with the results of the EVIDENT 2 and EVIDENT 3 studies, which analyzed the effectiveness of the use of the EVIDENT 2 and EVIDENT 3 applications to modify lifestyles in different subpopulations. The EVIDENT 2 study [48] for example, included the general population (833 subjects), aged 18 to 70 years. The objective was to assess the effect of adding an app for 3 months to the standard physical activity and heart-healthy eating advice on the modification of different measures of adiposity at 3 and 12 months after the intervention. In the global analysis, no differences were found in the body mass index. However, in the analysis by sex in the IG versus the CG at 12 months, a −0.67 cm difference in waist circumference was found. In addition, in 204 people between 25 and 70 years of age with type 2 diabetes mellitus, using the application of the EVIDENT 2 study in the intervention, beneficial results were obtained in the IG, both in greater adherence to the Mediterranean diet and increasing physical activity [49,50]. Similarly, analyzing the data of the 650 subjects included in the EVIDENT 3 study [51], the low-intensity intervention showed benefits on weight loss in the IG at 3 months (differences of −0.76 kg), on waist circumference at 3 months (differences of −0.76 cm) and in BMI at 3 months (differences of −0.30 points) with respect to CG. Finally, brief advice showed positive changes in macronutrients and in the intake of certain food groups. In addition, the intervention group reduced the intake of cholesterol (differences of −30 mg/dL, at 12 months) and full-fat dairy products and increased the intake of whole-grain bread and whole-grain cereals [52].

Moreover, the negative correlation between days of app use and the changes in arterial stiffness measures at 3 and 12 months indicate the importance of app adherence in these interventions.

On the other hand, interventions analyzing the effect of the use of mobile technologies on lifestyles have concluded that interventions with mobile phones improve markers of physical activity and adiposity [22], increasing the consumption of fruits and vegetables between 2 and 4 servings per day, without affecting the total intake of calories and sugary drinks [53]. A recent meta-analysis concluded that frequent and sustained interventions are needed to achieve clinically significant weight loss of 5% [42]. Such interventions also have favorable effects on blood pressure [54], weight [55], diabetes mellitus [56]) and increasing treatment adherence [57] while other clinical trials with more specific interventions have shown that a Mediterranean-style diet [58], or physical aerobic exercise [59] improve arterial stiffness.

Conversely, there are few studies evidencing the effect of the use of mHealth on arterial stiffness and other parameters derived from the pulse wave. Among them, the EVIDENT 2 study [32] with 597 subjects analyzed the effects of an intervention that included a brief lifestyle advice session together with the use of the EVIDENT 2 application on different arterial stiffness and pulse wave parameters; in the IG, a decrease in AIx75 was found at 3 months (−4.9%, 95% CI: −7.7 to −2.1) and 12 months (−3.9%, 95% CI: −6.8 to −1.0), although no differences were found in the other parameters analyzed.

These results indicate the need to analyses the characteristics of the Smartphone application itself. Thus, it is important to consider that the EVIDENT Smartphone application is an analytical application which, through the recorded information, generates information to allow the user to modify those lifestyle aspects in which he/she is failing to attain the goals set. Nevertheless, in the development of the App, motivational aspects were not taken into account, such as the social and affective frameworks, which seem to improve the results [60]. Furthermore, the optimal time of use of these interventions is not clear, and the long-term effect becomes dispersed with the passing of time, which, according to Afshin et al. [22], could be explained by the decrease of adherence to the App in time. Moreover, when the analysis was performed by age groups, differences were found between sexes [44,48], which suggests that it could be possible to identify the personal characteristics of those who would benefit to a greater extent from interventions based on new technologies, within the framework of health promoting activities. Thus, is manifested in the review of Afshin et al. [22], who highlighted the need for a more thorough evaluation of the efficacy, sustainability and usefulness of these interventions in different population groups. Therefore, some questions remain unanswered about the efficacy of conventional technologies and how to use them to modify physical activity and diet in adults who wish to lose weight, especially in the long term. Consequently, there is still a lack of conclusive evidence on the effectiveness of interventions that include apps and activity-tracking bands in the improvement of lifestyles. 

Nevertheless, the most notable results of this study are that, while clear effects were not achieved with the intervention, benefits were obtained on some of the measures analyzed with the standard lifestyle advice based on physical exercise and Mediterranean diet, even counteracting increases that occur with age. This suggests that brief-intensity multicomponent interventions are probably not suitable for producing effects on arterial stiffness and pulse wave-based parameters. New trials with more subjects are therefore necessary to confirm these findings.

### Strength and Limitations

The main strength of this study is the fact that it used a multicomponent intervention based on a Smartphone app and analysed the effect of this app on parameters that had not been studied to date, evaluating the short- and long-term effect, with good adherence to the diet app (median of 78 days over 90 days).

This study also has several limitations. Firstly, the nature of the intervention did not allow for the blinding of the subjects, which may affect the results, although some of the data indicate that this measure is not as important as was previously thought [61]. Secondly, although the subjects were requested not to use any other mHealth tool during the study, this could not be guaranteed. Thirdly, the duration of the intervention (3 months) may have not been enough to obtain greater changes, since some changes observed at 12 months suggest the importance of carrying out a follow-up of the effect in longer periods. Fourthly, the sample size, as well as the 15% loss of subjects at 12 months, may have reduced the statistical power of the study regarding the detection of a significant effect on the results between the two groups.

## 5. Conclusions

In sedentary adults with overweight and obesity, the multicomponent intervention (Smartphone application and an activity-tracking band) for 3 months did not modify arterial stiffness or the central hemodynamic parameters, compared to the control group. However, at 12 months, CG presented a decrease of cfPWV and cDBP, whereas IG showed a decrease of PAIx and ED and an increase of SEVR.

## Figures and Tables

**Figure 1 nutrients-14-04758-f001:**
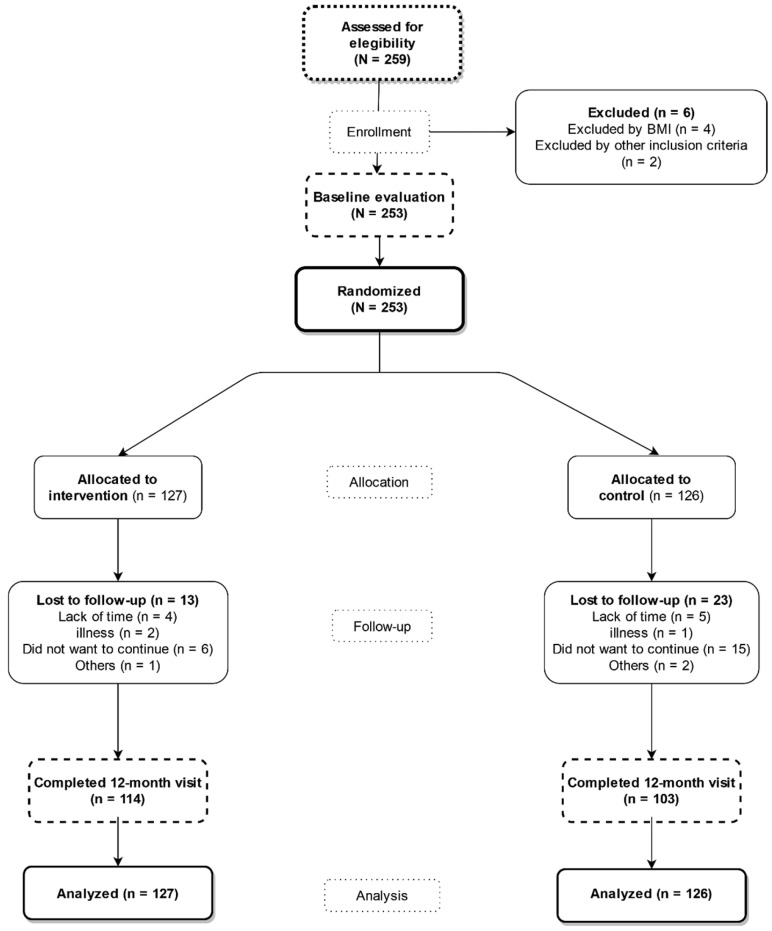
Study flowchart: recruitment of the subjects and completion of the study.

**Figure 2 nutrients-14-04758-f002:**
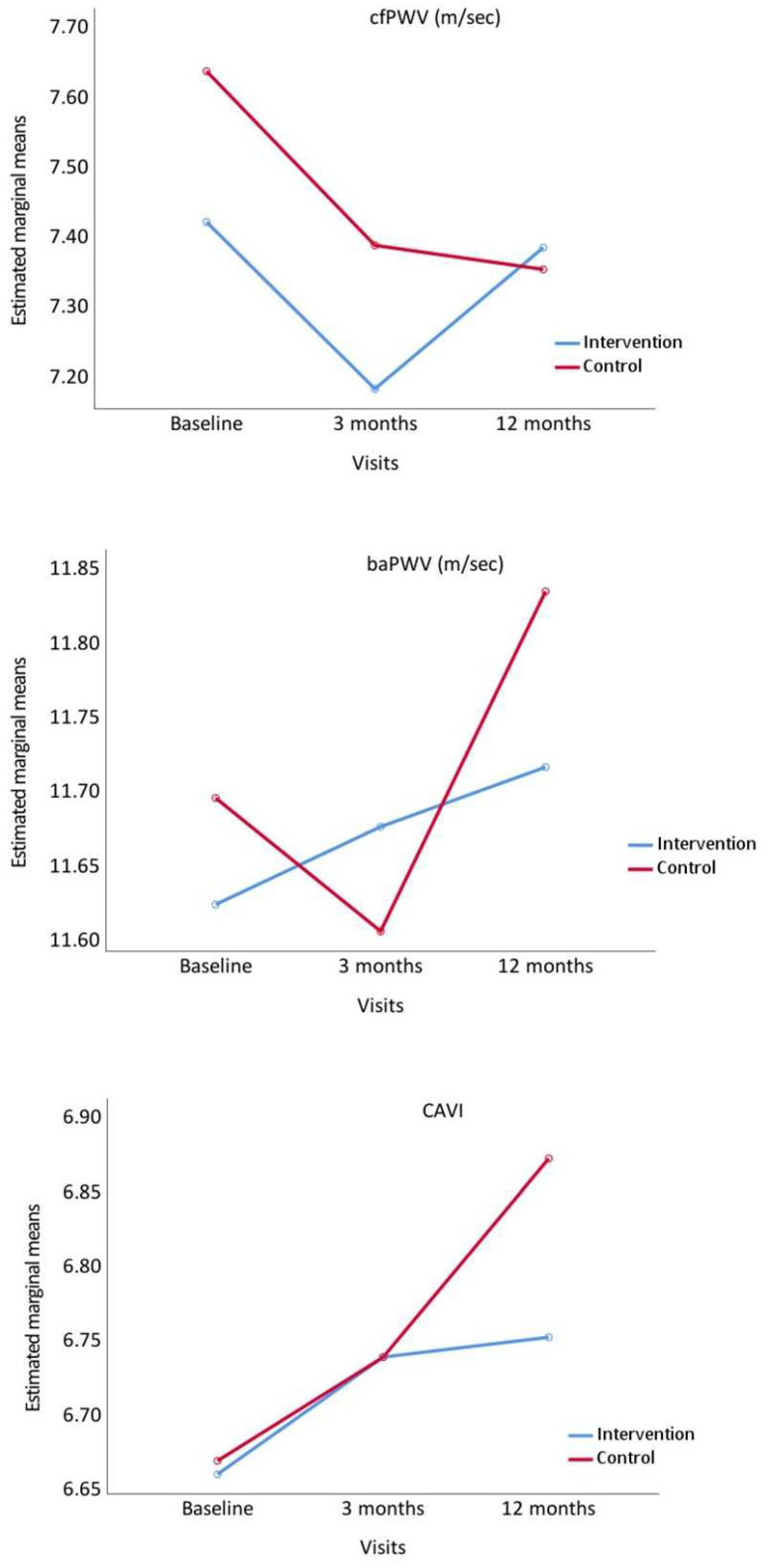
Evolution of carotid-femoral pulse wave velocity (cfPWV), brachial-ankle pulse wave velocity (baPWV) and Cardio Ankle Vascular Index (CAVI), from baseline to 3 and 12 months comparing the intervention and control groups. *p* value between groups: cfPWV, *p* = 0.475; baPWW, *p* = 0.851 and CAVI, *p* = 0.749.

**Figure 3 nutrients-14-04758-f003:**
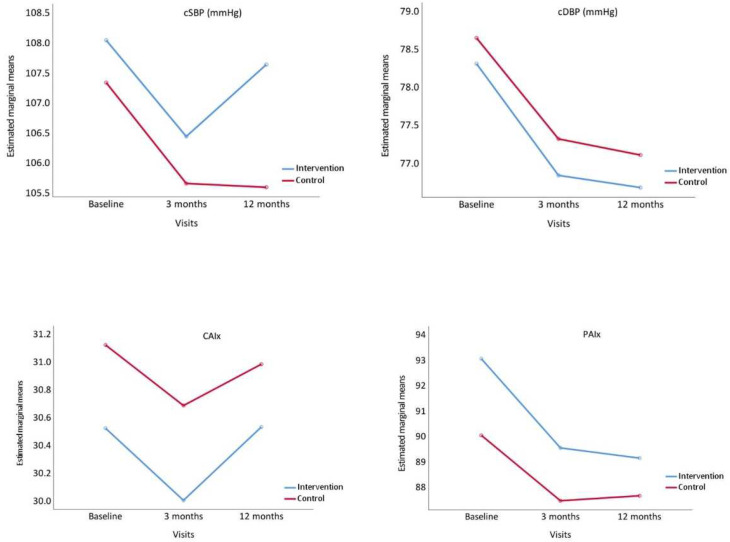
Evolution of central systolic blood pressure (cSBP), central diastolic blood pressure (cDBP), Central Augmentation Index, (CAIx) and Peripheral Augmentation Index (PAIx), from baseline to 3 and 12 months comparing the intervention and control groups. *p* value between groups: cSBP, *p* = 0.493; cDBP, *p* = 0.737, CAIx, *p* = 0.691 and PAIx, *p* = 0.375.

**Figure 4 nutrients-14-04758-f004:**
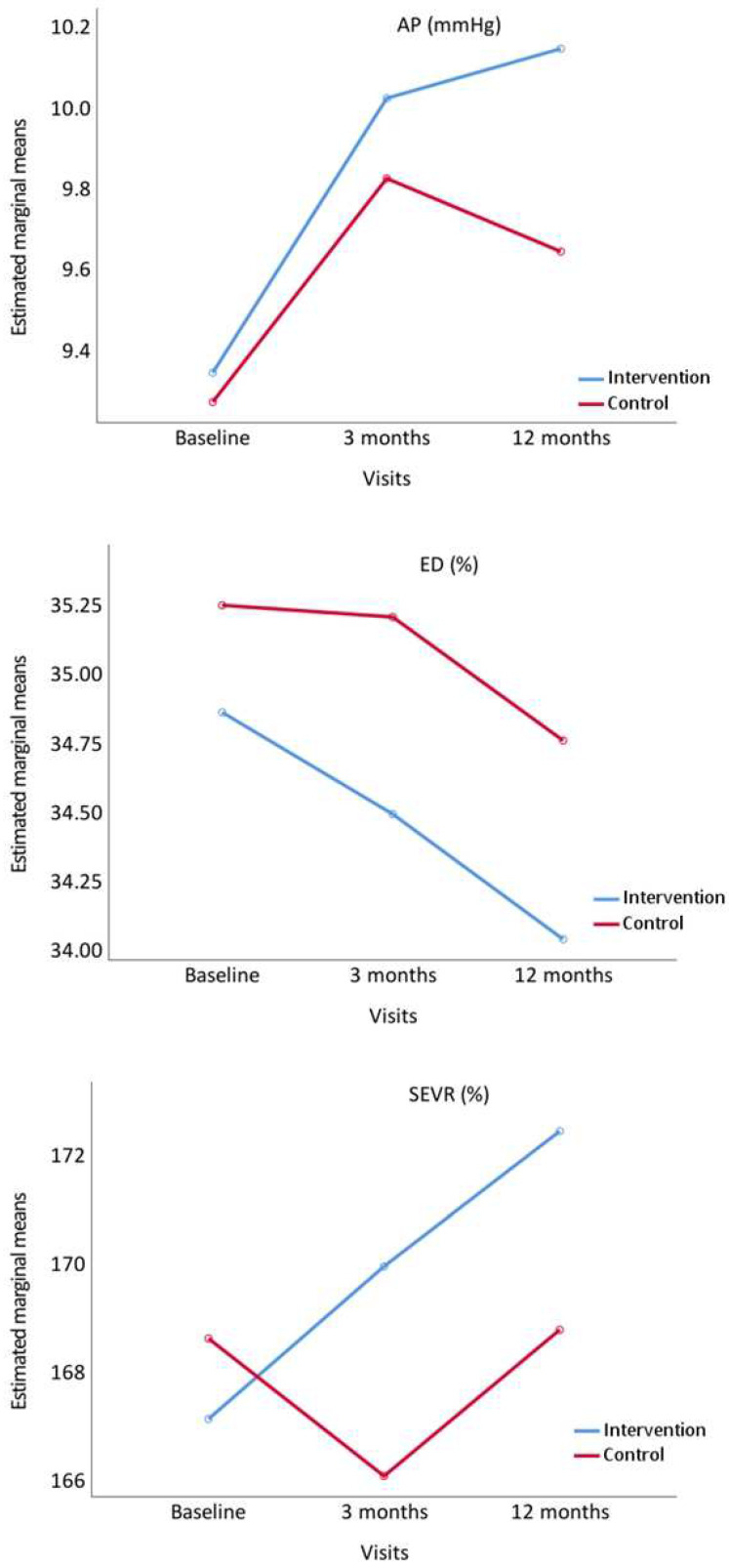
Evolution of augmentation pressure (AP), ejection duration ratio (ED%), and subendocardial viability ratio (SEVR%), from baseline to 3 and 12 months comparing the intervention and control groups. *p* value between groups: AP, *p* = 0.741; ED%, *p* = 0.276 and SEVR%, *p* = 0.583.

**Table 1 nutrients-14-04758-t001:** Baseline characteristics of study subjects.

Characteristics	Intervention Group (*n* = 127)	Control Group (*n* = 126)	*p* Value
**Age, mean years (SD)**	47.96 (9.95)	47.78 (9.62)	0.878
**Sex, n (%)**			0.413
Men	42 (33.1)	35 (27.8)	
Women	85 (66.9)	91 (72.2)	
**Smoking status, n (%)**			0.899
Non smoker	48 (37.8)	50 (39.7)	
Smoker	26 (20.5)	27 (21.4)	
Former smoker	53 (41.7)	49 (38.9)	
**Clinical variables, mean (SD)**			
BMI (kg/m^2^)	32,71 (3.27)	33.01 (3.31)	0.460
Systolic blood pressure (mmHg)	116.13 (14.50)	114.66 (14.14)	0.414
Diastolic blood pressure (mmHg)	79.29 (9.96)	79.10 (9.65)	0.876
Heart rate (bpm)	67.51 (9.01)	69.82 (10.42	0.060
Total Cholesterol (mg/dL)	194.61 (33.57)	190. 99 (33.91)	0.395
HDL Cholesterol (mg/dL)	49.89 (12.37	52.14 (11.72)	0.139
LDL Cholesterol (mg/dL)	118.79 (28.26	115.42 (29.99)	0.360
Triglicerides (mg/dL)	128.95 (76.85)	117.26 (58.02)	0.174
Glycaemia (mg/dL)	90.92 (12.22)	91.61 (19.49)	0.737
HbA1c (%)	5.46 (0.36)	5.49 (0.53)	0.536
Cardiovascular Risk (%)	5.89 (5.18)	6.07 (6.83)	0.807
**Chronic diseases, n (%)**			
Hypertension	32 (25.2)	36 (28.6)	0.322
Dyslipidemia	28 (22.0)	26 (20.8)	0.465
Diabetes Mellitus	1 (0.8)	3 (2.4)	0.308
**Medication use, n(%)**			
Antihypertensive drugs	17 (13.4)	23 (18.3)	0.187
Lipid-lowering drugs	22 (17.3)	19 (15.1)	0.377

BMI, body mass index; bmp, beats per minute; LDL—cholesterol, low-density lipoprotein cholesterol; HDL cholesterol, high-density lipoprotein cholesterol; HbA1c, glycosylated hemoglobin. Categorical variables are expressed as n (%) and continuous variables as mean ± standard deviation.

**Table 2 nutrients-14-04758-t002:** Effect of the mobile health intervention on arterial stiffness.

Parameters	Intervention Group (*n* = 127)		Control Group (*n*= 126)		Net Difference	
	Values	*p* Value	Values	*p* Value	Values	*p* Value ^a^
**cfPWV, m/s**						
Baseline, mean (SD)	7.42 (1.38)	N/A ^b^	7.64 (1.60)	N/A^b^	−0.22 (−0.60 to 0.15)	0.244
3-month change, mean difference (95% CI)	−0.18 (−0.38 to 0.02)	0.070	−0.28 (−0.54 to −0.02)	0.035	0.10 (−0.23 to 0.42)	0.558
12-month change, mean difference (95% CI)	−0.04 (−0.24 to 0.16)	0.702	−0.30 (−0.54 to −0.05)	0.018	0.26 (−0.05 to 0.57)	0.104
**baPWV, m/s**						
Baseline, mean (SD)	11.60 (1.44)	N/A ^b^	11.77 (1.73)	N/A ^b^	−0.17 (−0.56 to 0.22)	0.396
3-month change, mean difference (95% CI)	0.07 (−0.88 to 0.22)	0.342	−0.07 (−0.23 to 0.08)	0.338	0.15 (−0.07 to 0.37)	0.179
12-month change, mean difference (95% CI)	0.10 (−0.04 to 0.23)	0.160	0.13 (−0.06 to 0.33)	0.185	−0.03 (−0.27 to 0.21)	0.805
**CAVI**						
Baseline, mean (SD)	6.70 (1.02)	N/A ^b^	6.70 (1.20)	N/A ^b^	0.01 (−0.27 to 0.27)	0.992
3-month change, mean difference (95% CI)	0.07 (−0.07 to 0.20)	0.321	−0.09 (−0.07 to 0.25)	0.271	−0.02 (−0.23 to 0.19)	0.833
12-month change, mean difference (95% CI)	0.09 (−0.04 to 0.22)	0.183	0.22 (0.05 to 0.39)	0.012	−0.13 (−0.34 to 0.19)	0.243

cfPWV, carotid-femoral pulse wave velocity; baPWV, brachial-ankle pulse wave velocity; CAVI, cardio-ankle vascular index. ^a^ *p* value by analysis of variance; ^b^ N/A: not applicable.

**Table 3 nutrients-14-04758-t003:** Effect of the mobile health intervention on pulsatile hemodynamics.

Parameters	Intervention Group (*n* = 127)		Control Group (*n*= 126)		Net Difference	
	Values	*p* Value	Values	*p* Value	Values	*p* Value ^a^
**cSBP (mmHg)**						
Baseline, mean (SD)	108.17 (13.69)	N/A ^b^	107.18 (13.11)	N/A ^b^	0.99 (−3.34 to 4.32)	0.559
3-month change, mean difference (95% CI)	−1.43 (−3.30 to 0.44)	0.134	−1.93 (−3.52 to −0.35)	0.018	0.51 (−1.94 to 2.95)	0.688
12-month change, mean difference (95% CI)	-0.41 (-2.48 to 1.67)	0.699	−1.90 (−3.94 to 0.15)	0.069	1.49 (−1.41 to 4.39)	0.313
**cDBP (mmHg)**						
Baseline, mean (SD)	78.81 (9.84)	N/A ^b^	78.10 (11.87)	N/A ^b^	0.714 (−2.00 to 3.42)	0.605
3-month change, mean difference (95% CI)	−1.37 (−2.76 to 0.02)	0.053	−1.06 (−2.82 to 0.69)	0.232	−0.31 (−2.51 to 1.90)	0.785
12-month change, mean difference (95% CI)	−1.63 (−3.36 to 0.09)	0.064	−1.64 (−3.19 to −0.10)	0.037	0.01 (−2.29 to 2.32)	0.991
**CAIx**						
Baseline, mean (SD)	30.81 (12.14)	N/A ^b^	30.42 (11.79)	N/A ^b^	0.39 (−2.58 to 3.37)	0.795
3-month change, mean difference (95% CI)	−0.85 (−2.66 to 0.96)	0.353	−0.36 (−2.33 to 1.60)	0.715	−0.49 (−3.15 to 2.17)	0.717
12-month change, mean difference (95% CI)	0.01 (−2.07 to 2.09)	0.993	−0.05 (−2.16 to 2.06)	0.961	0.006 (−2.89 to 3.01)	0.967
PAIx						
Baseline, mean (SD)	92.19 (20.82)	N/A ^b^	89.03 (18.62)	N/A ^b^	3.16 (−1.84 to 8.17)	0.214
3-month change, mean difference (95% CI)	−2.73 (−5.95 to 0.49)	0.096	−1.56 (−4.52 to 1.40)	0.297	−1.17 (−5.55 to 3.18)	0.597
12-month change, mean difference (95% CI)	−3.60 (−7.22 to 0.00)	0.050	−2.09 (−5.26 to 1.07)	0.193	−1.51 (−6.29 to 3.28)	0.538
**AP (mmHg)**						
Baseline, mean (SD)	9.31 (4.99)	N/A ^b^	8.91 (4.80)	N/A ^b^	0.40 (−0.84 to 1.61)	0.521
3-month change, mean difference (95% CI)	0.43 (−1.04 to 1.92)	0.558	0.42 (−1.18 to 2.02)	0.604	0.02 (−2.15 to 2.19)	0.987
12-month change, mean difference (95% CI)	0.80 (−0.79 to 2.39)	0.320	0.34 (−1.41 to 2.10)	0.699	0.46 (−1.89 to 2.81)	0.701
**ED (%)**						
Baseline, mean (SD)	35.02 (4.18)	N/A ^b^	35.37 (6.39)	N/A ^b^	0.35 (−1.70 to 0.99)	0.795
3-month change, mean difference (95% CI)	−0.52 (−1.12 to 0.07)	0.087	−0.04 (−1.23 to 1.16)	0.951	−0.49 (−1.76 to 0.79)	0.454
12-month change, mean difference (95% CI)	−0.82 (−1.36 to −0.27)	0.003	−0.54 (−1.72 to 0.64)	0.367	−0.28 (−1.54 to 0.98)	0.662
**SEVR (%)**						
Baseline, mean (SD)	167.48 (26.63)	N/A ^b^	167.06 (26.25)	N/A ^b^	0.42 (−6.15 to 6.99)	0.900
3-month change, mean difference (95% CI)	2.90 (−0.87 to 6.67)	0.130	−2.07 −5.95 to 1.80)	0.291	4.98 (−0.42 to 10.38)	0.070
12-month change, mean difference (95% CI)	5.31 (1.18 to 9.44)	0.012	0.56 (−4.96 to 6.08)	0.840	4.75 (−2.11 to 11.60)	0.168

cSBP, central systolic blood pressure; cDBP, central diastolic blood pressure; CAIx, Central Augmentation Index; PAIx, Peripheral Augmentation Index; AP, Augmented pressure; ED%, ejection duration ratio; SEVR%, subendocardial viability ratio. ^a^ *p* value by analysis of variance; ^b^ N/A: not applicable.

## Data Availability

The datasets used and/or analyzed during the current study are available from the corresponding author on reasonable request.

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
