# Peer review of "Long-Term Effectiveness of a Smartphone App and a Smart Band on Arterial Stiffness and Central Hemodynamic Parameters in a Population with Overweight and Obesity (Evident 3 Study): Randomised Controlled Trial"

_nutrients, 2022, doi:10.3390/nu14224758_

Round 1
Reviewer 1 Report
Figures lack clarity. Procedures implemented can be explained a little more
Author Response
The answers are in the attached file.
Reviewer 2 Report
Introduction is complete and well written. It explains the variables and outcomes analyzed in the research paper. I suggest to state the objectives of the study in a clearer way, at the end of Introduction. However I believe that Introduction needs to present the rationale of your study. For example, what is the reason why you allowed to use the app only for three months?
Lines 304-305, 311-312: I believe you should discuss these findings.
In the comparison of the two groups, it would be interesting to know if there were a signficant difference in other outcomes, that are related to cardiologic health, for example weight loss. I understand that other results are presented in another paper, but I think it is relevant for the overall interpretation of results. You state something at lines 378 and following but it is not clear if the groups are the same.
Also I believe you should deepen the literature review in terms of interventional studies on lifestyle which managed in reducing the selected outcomes. In particular, studies which involves the use of mobile technologies. Is there any evidence that interventions in terms of changing lifestyles can bring to improvements of the indexes you choose in 3-12 months?
Indeed, I found interesting the fact that in IG, those who used more the app showed better results, this is something that I think you should take into consideration also in Discussion.
Author Response
The answers are in the attached file.

Reviewer 3 Report
With the presents advances in technology new wearable sensors are developed which are integrated with mobile phones through a mobile application. The most mobile applications are focused on monitoring the physical activity and calories intake. The authors use such a mobile application, previously developed, in a study on arterial stiffness and central hemodynamic parameters.
1. What kind of mathematical transformation is used to estimate the aortic pulse wave and central arterial pressure?
2. More details about the mobile application should be included. What kind of foods and meals are present in the app? Are the participants restricted just to consume the meals/foods included in the app?
3. What does the activity- tracking band measures and how is it transmitted and stored in the app?
4. A more detailed discussion should be introduced with respect with the results not only a description of the graphs which are very clear: e.g. “ Figure 2 shows the evolution of the measurements of stiffness analysed throughout the study. cfPWV decreased at 3 months in the two groups, although the decreasing tendency remained in CG at 12 months, while this parameter increased in IG. baPWV and CAVI increased at 12 months in the two groups, presenting a greater increasing tendency in CG.” The explanation that this behavior is in accordance with the results in a previous study, is not sufficient.
5. What are the functionalities of the mobile app and how is the input data (meals/food intake, physical activity etc) analyzed and processed by the app.
6. The discussion consists on remarks about a previous study that the authors implemented. It should be more focused on the results presented in the current study. What is the added value of the results presented in this study compared to previous one?
7. The quality of figure 1 must be improved.
Author Response
The answers are in the attached file.

Round 2
Reviewer 3 Report
The authors have responded to my comments.
Moderate English changes are still required.
Author Response
The manuscript has been edited by Robert Raabe BA, MSc.
We attach certificate in attached file
